# Scanning Distance Influence on the Intraoral Scanning Accuracy—An In Vitro Study

**DOI:** 10.3390/ma15093061

**Published:** 2022-04-22

**Authors:** Raul Nicolae Rotar, Andrei Bogdan Faur, Daniel Pop, Anca Jivanescu

**Affiliations:** 1Department of Prosthodontics, University of Medicine and Pharmacy Victor Babes, B-dul Revolutiei 1989, No. 9, 300580 Timisoara, Romania; rotarraul3@yahoo.com (R.N.R.); dr.popdan@gmail.com (D.P.); ajivanescu@yahoo.com (A.J.); 2TADERP Research Center, 300041 Timisoara, Romania

**Keywords:** intraoral scanner, CAD/CAM, accuracy, scanning distance

## Abstract

Intraoral scanners (IOS) have reached a point where their impact in the dental office cannot be denied. The distance between the tip of the IOS and the preparation may have implications on the accuracy of the digital model. The objective of this study was to evaluate the differences in accuracy between digital impressions in the scenario of different scanning distances. Twenty consecutive scans were performed at five predetermined distances: 5 mm, 10 mm, 15 mm, 20 mm and 23 mm by a single operator. The scanning distance of 10 mm displayed the best accuracy with an overall trueness value of 23.05 μm and precision value of 4.2 μm. The drawn conclusion was that increased scanning distances can decrease the accuracy of a digital impression.

## 1. Introduction

Intraoral scanners (IOS) have reached a point where their impact in the dental office cannot be denied. The advantages that these devices bring to the table when compared to the conventional ways of recording the intraoral structures are represented by the working speed, the comfort for the patient, intraoperative assessment of the preparations, elimination of cross infections and distortion of the impressions and the possibility of indefinite storage of digital models [1,2,3]. When combined with a CAD/CAM (Computer-Aided Design/Computer-Aided Manufacturing) software, the applications of IOSs range from single tooth prosthetic restorations [4,5,6], fixed partial dentures [7,8], implant supported restorations and even removable dentures [9,10]. However, these systems have a number of limitations which at the moment, in some cases, can restrict their use in all clinical scenarios. For a predictable treatment, IOSs require a clear identification of finish line geometry, deep vertical preparations may not allow a sufficient penetration of the light for a correct digital reconstruction, presence of humidity can distort the scanned surface as well as the ambient lighting condition [11,12,13,14].

The assessment of the scanning accuracy of an IOS is made by evaluating two parameters: trueness and precision (ISO 5725). Trueness indicates the degree in which the analogue model is reproduced by the digital model. The precision shows the repeatability of a measurement therefore it compares the degree in which multiple virtual models correspond with each other [15,16].

An accurate impression is essential for the predictability and longevity of a prosthetic restoration. Even if IOSs with the digital impression techniques aim to eliminate the disadvantages of conventional impressions, there are still some situations that require increased attention from the clinician. Failing to recognize these factors may lead to errors of the digital model which consequently may lead to inadequate restorations. There are a number of studies that investigate the influence of these factors regarding the accuracy of digital impressions. The scanning strategy and the exposure time of a particular area are important aspects of intraoral scanning since following a correct scanning pattern combined with lower scanning times can improve the accuracy of digital impressions [17]. The presence of fluids on the recorded area may also lead to inaccuracies due to the alteration of optical properties of dental structures [18]. Another factor that may lead to inaccuracies is related to the distance of scanning. During the scanning procedure, it is extremely difficult for an operator to maintain a constant spacing between the scanning tip and the recorded teeth or soft tissues, therefore is important to know if certain scanning distances provide better accuracy results than other.

The objective of this study was to evaluate the differences in accuracy between digital impressions in the scenario of different scanning distances.

## 2. Materials and Methods

For this study, a typodont (AG-3; Frasaco GmbH, Tettnang, Germany) was selected. An all-ceramic overlay preparation with a rounded finish line, 1 mm of axial reduction and 1.5 mm of occlusal reduction were conducted on a first upper right molar of the typodont (Figure 1).

Twenty consecutive scans were performed at five predetermined distances: 5 mm, 10 mm, 15 mm, 20 mm and 23 mm by a single operator. The IOS used in this study was i700. i700 (Medit, Seoul, Korea) is the latest intraoral scanner from Medit, improving most of the features presented by Medit i500 (Medit, Seoul, Korea). The principle of 3D in Motion Video Technology/3D Full Color Streaming Capture is used for data acquisition and can export the files in open formats (OBJ, PLY and STL) [19]. To maintain the desired distance between the scanner’s tip and the overlay preparation, the IOS was attached to a rigid articulated arm. Before each scan, the distance was measured with a digital caliper (Figure 2). Calibration was performed according to the manufacturer’s instructions. The parallelism of the scanner to the occlusal plane was adjusted after each scan.

The scanning was performed only on the occlusal surface of the preparation, and for the mesh alignment, the occlusal surfaces of the second molar and premolar were recorded as well. The scanning path was identical for all the distances with a total scanning time of 20 s per scan. The typodont was moved using the following pattern: starting from the occlusal surface of the first molar, continuing to the occlusal of the second molar, the second premolar and ending on the initial starting point. Movement of the typodont was performed in a linear direction and at all times rested on the flat surface.

In order to obtain the reference digital model, Medit T500 desktop dental scanner was used. Medit T500 (Seoul, Korea) is a desktop dental scanner providing an accuracy of <7 μm, being based on the principle of phase-shifting optical triangulation, and being equipped with a blue LED light source [20].

In order to process the scanned data, Geomagic Control X (Version:16.0.2.16496, 3D Systems, Wilsonville, OR, USA), a complete metrology grade, quality control software, was uesed. 

The reference data was uploaded into the software. The occlusal surface area of the first molar was isolated from the rest of the mesh as the area of interest. The isolating procedure respected the margins of the preparations and was done manually inside the software. For consistency, the area of interest remained unchanged for all the following comparisons, not being necessary to retrace the margins of the area of interest as the reference data remains the same and only the measured data is being swapped. The intraoral scanner mesh was uploaded into the software as measured data to be compared with the reference data. The isolated area of interest was the only area analyzed between the meshes, excluding the remaining occlusal areas from the molar and the premolar.

The trueness data was obtained by executing the “initial alignment” function to superimpose the intraoral scanner mesh over the reference mesh followed by the “best fit alignment” (Figure 3). In order to analyze the deviation between the reference and measured data, the “3D Compare” function was used by projecting all paired points onto the reference data. A color-coded map was rendered displaying the deviation patterns of the investigated surfaces. The color-coded map was set to display deviations between ±50 μm (Figure 4). In order to obtain the precision values, each intraoral scanner mesh was compared with all the other meshes within its group following the same exact same steps.

MedCalc statistical software was used to conduct the statistical analysis by uploading the standard deviation data obtained from the metrology software. The Kolmogorov Smirnov test resulted that the data for both trueness and precision was non-parametric. Consequently, Kruskal-Wallis test was used to further analyze the data. The level of significance was set to α = 0.05.

## 3. Results

Table 1 presents the median values for the trueness and precision measurements.

The Kruskal-Wallis test indicated that there are significant relevant differences between the groups for both trueness and precision data (*p* < 0.001). The scanning distance of 10 mm displayed the best accuracy with an overall trueness value of 23.05 μm and precision of 4.2 μm.

Table 1 shows that both trueness and precision values increase with the increase of the scanning distance, meaning that the overall accuracy is lower. The lowest accuracy was found at the scanning distance of 23 mm. Also, at 5 mm the accuracy was lower than at 10 mm.

## 4. Discussion

An accurate impression is one of the most important steps when referring to predictable treatments and long-lasting restorations. When used correctly, the IOS can provide invaluable feedback to the clinician, especially regarding the impression quality as well as the overall shape and geometry of preparation [21]. Even though the manufacturers provide for each brand of IOSs a set of guidelines regarding their proper use, these instructions may not or cannot be respected in all clinical scenarios [22].

The present study investigated the trueness and precision of an intraoral scanner (Medit i700) performing at five different scanning distances. The selected distances were based on possible clinical situations: 5 mm (when scanning a second or third molar where the interarch space forces the scanning tip position to be very close to the preparation) up to 23 mm (which can happen for the anterior teeth or in the case of a deep preparation combined with an increased scanning distance). We also tried to exceed the maximum scanning distance recommended by the manufacturer for this scanner to 25 mm, but achieving a usable scan was not possible. The first molar was chosen for the current study due to the larger occlusal surface when compared to the second and third molar since the scanning procedure was performed only from an occlusal direction. Additionally, the presence of both adjacent teeth was another reason since the occlusal surfaces of the second premolar and molar played an important role in the mesh alignment process. The influence of the operator regarding the scanning distance was minimized since all the scans were made with the scanner fixed at the predetermined height and with 20 s of scanning time per digital impression. We observed for the 23 mm scanning distance that the scanner required a longer scanning time to record the occlusal surfaces. Therefore, we applied this exposure time (20 s) to all the scanned distances in order to remove this variable from the current analysis’. The scanning procedure implied the use of a rigid articulated arm in order to maintain the desired distance between the scanner and the occlusal surfaces. This meant that the position of the scanner remained the same for each set distance.

There are a number of factors that can influence the accuracy of a digital impression. Wesemann et al. investigated the influence of ambient light in the scanning time and impression accuracy for six IOSs (TRIOS 3 (TRI), Cerec Omnicam (OC), iTero Element (ITE), CS 3600 (CS), Planmeca Emerald (EME), and GC Aadva (AAD)) and concluded that ambient light influenced both the accuracy and scanning time of IOSs [23]. The presence of humidity on the prepared tooth which changes the reflection and refraction of light on that surface, and the number of passes/scans over the same area [24]. Muller et al. evaluated the influence of the scanning strategy on the accuracy of a digital impression using three approaches (A, first buccal surfaces, return from occlusal-palatal; B, first occlusal-palatal, return buccal; C, S-type one-way) and concluded that the B strategy ensures he highest trueness and precision in full-arch scan [25].

The aim of this study was to evaluate only the ‘scanning distance factor’, without the influence of other possible clinical factors such as humidity or ambient lighting conditions. As a result, possible clinical factors such as humidity, patient movements or presence of blood, were not simulated, therefore they did not influence the final accuracy of the impressions. The ambient light intensity was measured and quantified using a digital lux meter (GM1010; Benetech, Palo Alto, CA, USA) with the measuring range of 0~200,000 and considered the most relevant and plausible as neon ambient light on a sunny day-measuring 1000 lux. No additional external light sources were used.

The present study has several limitations. Being an in vitro study, all clinical factors such as the presence of saliva, patient movements or the interference with the soft tissues, were not taken into consideration. Also we evaluated only one IOS, meaning that other brands of IOSs can generate different results. Therefore, additional studies are required including multiple IOSs to observe whether or not the scanning distance factor also influences their accuracy. The results of this study showed that the distance of the scanning tip is an influencing factor for the final accuracy of a digital impression. We observed that increasing the scanning distance decreases the accuracy of a scan, being more evident at the 20 mm and 23 mm range. One possible reason for having lower accuracy at the 5 mm distance can be related to the intensity of the light emitted by the scanner and the proximity of the recorded area, meaning that some points may become oversaturated and lead to alterations of the digital model. Going over the 15 mm distance, the decrease in accuracy can be attributed to the inability of IOS sensors to successfully capture all the reflected light form the scanned surface. In a clinical scenario, it is unlikely that the operator will maintain the scanning tip at only one set distance, therefore areas with lower accuracy caused by an increased scanning distance will be compensated by secondary passes following the scanning patterns. However, errors may appear when scanning deep placed implants with mesial and distal adjacent teeth that condition the scanning distance.

## 5. Conclusions

Based on the results of the present study, the following conclusion can be drawn:The distance between the tip of the IOS and the recorded surface can influence the accuracy of a digital impression.Close scanning distances (5 mm) or scanning distances that exceed 15 mm have a negative impact on the accuracy of the digital model.The distance of 10 mm between the scanning tip and the prepared area showed the best accuracy.

## Figures and Tables

**Figure 1 materials-15-03061-f001:**
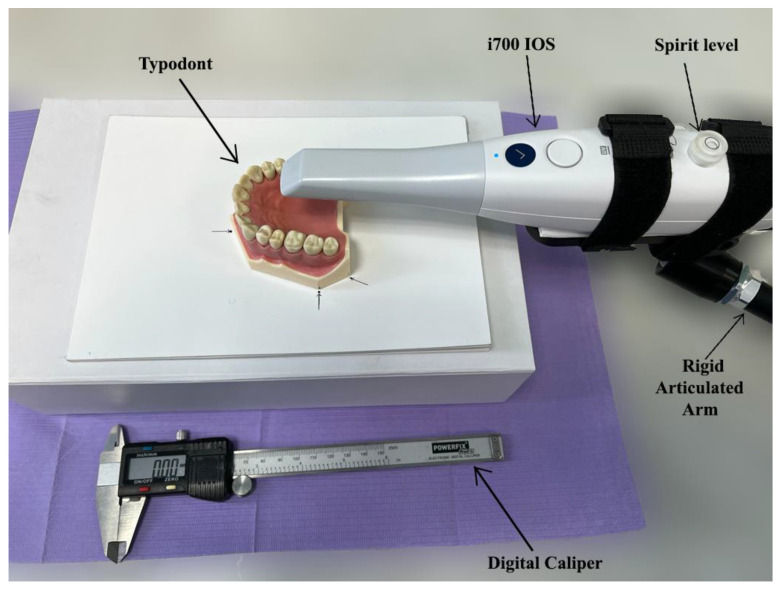
Visual description of the experiment setup and the instruments used in the process.

**Figure 2 materials-15-03061-f002:**
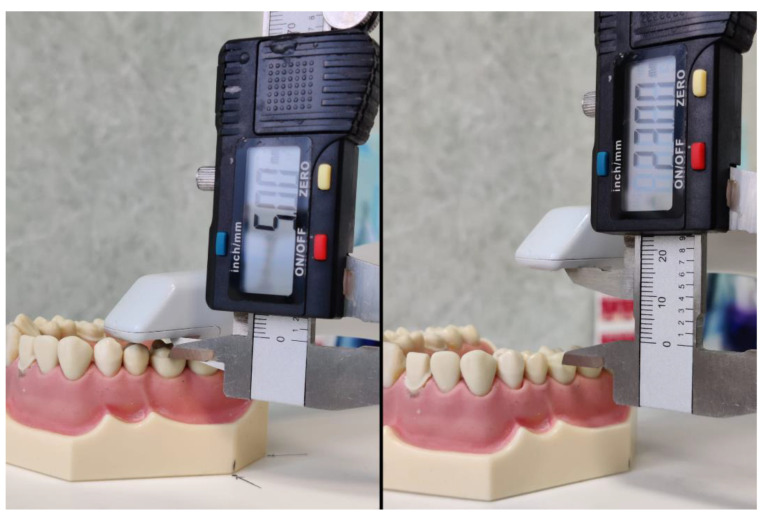
Distance calibration with the digital caliper.

**Figure 3 materials-15-03061-f003:**
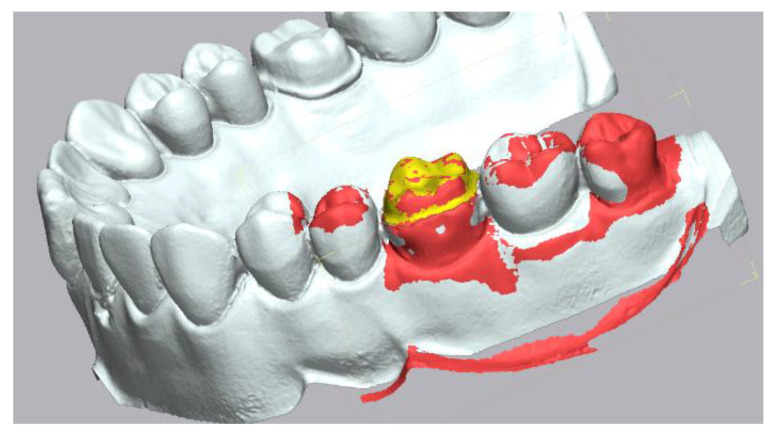
Alignment of the IOS mesh over the reference mesh. The isolated area of interest is displayed in yellow color.

**Figure 4 materials-15-03061-f004:**
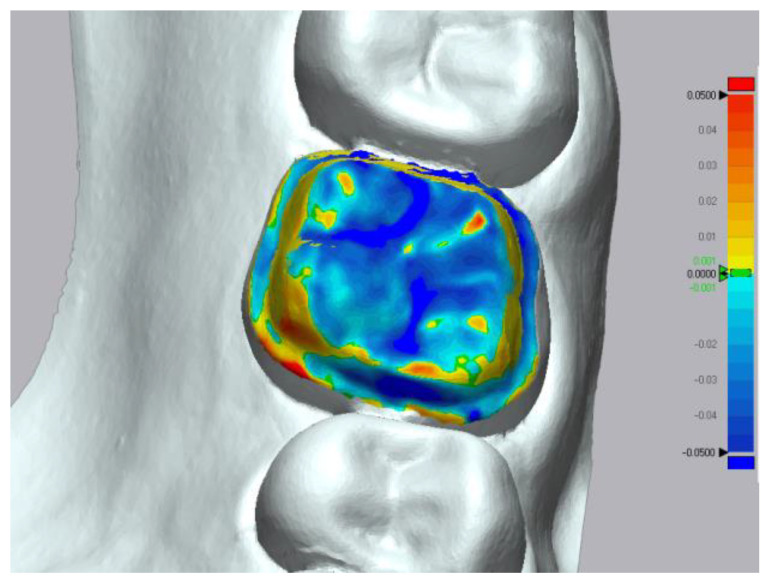
Color-coded map generated by the metrology software. Deviation is displayed in mm on the color scale with outward (red) and inward (blue) displacement.

**Table 1 materials-15-03061-t001:** Median values for trueness and precision data.

Scanning Distance	5 mm	10 mm	15 mm	20 mm	23 mm
Trueness-Median (IQR)	35.1 (5.2) μm	23.05 (1.05) μm	40.7 (10.9) μm	71.4 (2.3) μm	78.5 (1.5) μm
Precision-Median (IQR)	6.7 (1) μm	4.2 (2.4) μm	4.8 (0.5) μm	8.3 (2) μm	38.6 (15.1) μm

## Data Availability

All data is available upon request.

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
