# Peer review of "Scanning Distance Influence on the Intraoral Scanning Accuracy—An In Vitro Study"

_materials, 2022, doi:10.3390/ma15093061_

Round 1
Reviewer 1 Report
The present study investigated the influence of scanning distance on the trueness and precision of the IOS. The results clearly indicated that the 10-mm distance is the best for the scanning. The finding is of interest to dentists for use of IOS. I recommended this paper after a revision. My comments is listed as follows.
- Table 1 and Figs 5 and 6
The results in Table 1 duplicate those of Figs 5 and 6. These results should be consolidated into one table or two figures.
- IOS
This study used one brand of IOS (i700). On the other hand, the ability of IOSs may depend on the brands. Please discuss that influence of types of IOS on the trueness or precision.
- Ambient condition
In discussion section, the author mentioned that both ambient light and humidity may affect the scanning result. In this experiment, on the other hand, the scanning was performed using the typodont model. This situation differs from a real oral environment. Therefore, the author should discuss on this situation. In addition, please add the present conditions, brightness and humidity, around the typodont model.
- Limitation
Please mention the limitations of this study.
Reviewer 2 Report
This manuscript No. materials-1668643 titled “Scanning distance influence on the intraoral scanning accuracy-An in vitro study” was interesting. However, the novelty of this article was not shown in the paragraph, and we suggest the authors increase some paragraphs on the clinical values. In addition, some specific points were needed to be clarified.
Specific:
- Please add paragraphs to clarify all factors that affect the accuracy of the oral scan. Please emphasize the importance of scan distance, distance limitation, and why choose 5/10/15/20/23mm distance in this research.
- Why did the authors only choose the 1st molar for accuracy comparison? It is recommended for additional description in the paragraph on Materials and Methods.
- The results of this article only show the accuracy of ONE single tooth (the 1st molar) after twenty scans. The study’s applicable clinical value was expected to highlight whether the same or similar results could be obtained in other tooth positions.
- We suggest adding all included scanned teeth to compare the accuracy. In addition, the superimposition method and area in this study might affect the results, and the authors should prove or clarify the technique.
- Fig 1 needed to be re-photo. It showed the different scan positions from this study. By the way, the scanner position was not appropriate for clinical application.
- The scan path was not followed the clinical path, and please clarify whether the movement was from the model or the oral scanner.
- Why only choose the occlusal side to scan? The authors mentioned that the scan time was 20 seconds from 2ndpremolar to 2nd Therefore, the scan speed and frequency was needed to be clarified.
- The results point out that the distance of 10 mm between the scanning tip and the prepared area showed the best accuracy, but the authors did not explain why the accuracy of groups 5 mm and 15 mm was worse than 10 mm, and this explanation should be reinforced in the discussion.
- In addition, the authors used the model as a benchmark for measurement, and there is no difference in implant depth. Hence, it is not appropriate to include references (Refs. 21-23) to implant depth affecting accuracy in the discussion because they are not highly relevant to the topic of this study.
- The first conclusion seems easy to imagine and is not of high academic importance. Therefore, it is strongly suggested to rewrite the conclusion to make it more valuable.
- The reference format should follow the ”author guide of materials.”
Round 2
Reviewer 2 Report
- Please add paragraphs about the factors (scan strategy, head movement, scan speed, environment factors, operator skills, etc.) that affect the accuracy of the oral scan in introduction and emphasize the importance of the scan distance in introduction to enhance the research motivation and purpose.
- When citation literature, From lines 2 to 4 on page 5, “Weseman et al.…time of IOSs [13,21].”. Ref. 13 is not a study by Weseman et al., authors should mark it separately from Ref. 21. Please check all citations.
- The scanning path/procedure paragraph should be moved from discussion to materials and methods.
- The resolution of figure 1 and figure 2 is poor; please change the figures.
- The reference format needs to be rechecked carefully and follow the author’s guide.
